# A Molecular Dynamics Study of Heat Transfer Enhancement during Phase Change from a Nanoengineered Solid Surface

**A. K. M. Monjur Morshed [1], Muhammad Rubayat Bin Shahadat [2], Md. Rakibul Hasan Roni [1], Ahmed Shafkat Masnoon [3], Saif Al-Afsan Shamim [1] and Titan C. Paul [4],***

[1] Department of Mechanical Engineering, Bangladesh University of Engineering and Technology, Dhaka 1000, Bangladesh; monjur_morshed@me.buet.ac.bd (A.K.M.M.); rakibulhasan@me.buet.ac.bd (M.R.H.R.); shamimafsan@me.buet.ac.bd (S.A.-A.S.)

[2] Department of Mechanical Engineering, Michigan State University, East Lansing, MI 48824, USA; shahadat@msu.edu

[3] Power Grid Company of Bangladesh Limited, Dhaka 1212, Bangladesh; shafkat.masnoon@gmail.com

[4] Department of Mathematical Sciences, University of South Carolina Aiken, Aiken, SC 29801, USA

* Correspondence: titanp@usca.edu

**Abstract:** This study investigates the enhancement of the rate of evaporation from a nanoengineered solid surface using non-equilibrium molecular dynamics simulation. Four different types of surface modifications were introduced to examine the thermal transportation behavior. The surface modification includes: (1) transformation of surface wetting condition from hydrophobic to hydrophilic, (2) implementing nanostructures on the smooth surface, (3) cutting nano slots on the smooth surface and (4) introducing nano-level surface roughness. Evaporation behavior from the same effective surface area was also studied. The simulation domain consisted of three distinct zones: solid base wall made of copper, a few layers of liquid argon, and a vapor zone made of argon. All the nano-level surface modifications were introduced on the solid base surface. The few layers of liquid argon representing the liquid zone of the domain take heat from the solid surface and get evaporated. Outside this solid and liquid zone, there is argon vapor. The simulation began at the initial time t = 0 ns and then was allowed to reach equilibrium. Immediately after equilibrium was achieved on all three-phase systems, the temperature of the solid wall was raised to a higher value. In this way, thermal transportation from the solid wall to liquid argon was established. As the temperature of the solid wall was high enough, the liquid argon tended to evaporate. From the simulation results, it is observed that during the transformation from hydrophobic to hydrophilic conditions, enhancement of evaporation takes place due to the improvement of thermal transportation behavior. At the nanostructure surface, the active nucleation sites and effective surface area increase which results in evaporation enhancement. With nano slots and nano-level surface roughness, the rate of evaporation increases due to the increase of solid-liquid contact area and effective surface area.

**Keywords:** molecular dynamics simulation; phase change heat transfer; wettability; nano slot; surface roughness

## 1. Introduction

Molecular dynamics simulation had been used by a number of researchers to investigate microscopic phenomena of evaporation and boiling of an ultra-thin liquid layer over a solid surface. Due to its increasing importance in the cooling of electronic devices [1], energy storage [2], laser surgery [3], and laser stream cleaning [4] researchers had been keen to examine the nanoscale phase transition phenomena. Different studies have shown that heat transfer rate and boiling efficiency depend on the nature of the solid surfaces [5,6]. Researchers have also found that a nanoengineered surface shows a significant enhancement in heat transfer as well as reduced thermal interface resistance [7–9]. While surface modification promotes mixing and boosts up thermal performance, it also introduces an

increased pressure drop across the modified surface [10,11]. Researchers are keen to understand the nanoscale behavior of thin layer boiling as it can help meet the high demand for electronic cooling in the manufacturing industry. For the last two decades, molecular dynamics (MD) simulation has been a popular tool to study the nature of phase transition phenomena at the atomic level. MD simulation has opened a new era for scientists to investigate heat transfer at a microscale. Heat transfer through nanomaterials, capillary flow, and thin layer fluid, can easily be analyzed by MD simulation.

Morshed et al. [12] carried out non-equilibrium molecular dynamics (NEMD) research to investigate the size effect of nanostructures on boiling using platinum as a solid surface. Similar research had also been done by Wang and Zhang for the case of an aluminum surface at an elevated solid surface temperature [13]. Shavik and Hasan used MD simulation to study the nature of surface wettability on nanoscale boiling [14]. These studies help understand the difference between macroscale and microscale boiling, thermal resistance at the solid-liquid interface, and the effect of modified surface on boiling characteristics. Although many previous studies were conducted in this field, there are few studies on the effects of nano slots and nano level surface roughness. In addition, few studies can be found on evaporation behavior when keeping the effective surface area the same. There was always a need to bring together all these surface modifications in a single paper for future researchers which was our main motivation.

Argon was considered as a very compatible material for these numerical analyses. Interatomic interactions among argon molecules can easily be modeled by using Lennard-Jones (LJ) potential. To investigate phase transition phenomena an argon model can be simulated at a shorter time because of its monatomic structure. Previous studies have shown the compatibility of using argon as a liquid or a vapor. In recent years, the simulation model used for argon has also been tried for real materials and the results were found to be satisfactory [15].

In this study, the boiling of a thin layer of liquid argon over a solid copper surface is examined through steady NEMD simulations. The liquid is modeled by LJ potential whereas the interaction between solid and liquid is modeled by modified LJ potential based on Lorentz–Berthelot combining rules. The boiling nature of a thin liquid layer is observed for different physical properties of the surface. The solid surface is changed from hydrophilic to hydrophobic and from smooth surface to nanodot decorated surface. The boiling pattern over a solid surface with slotting and introducing roughness on the solid surface is also observed. Our literature review in the field suggests that this is the first work combining all the possible modifications of the solid layer concerning the ultra-thin liquid film boiling.

## 2. Methodology

The molecular system consisted of a solid copper wall, liquid argon layer, and vapor argon layer in a cuboid shape whose dimensions were $6.08 \times 32.42 \times 6.08$ nm$^3$. At the bottom of the simulation domain, there were six monolayers of copper atoms that act as a solid wall. Above the solid wall, there were eleven monolayers of liquid argon atoms, and the rest of the simulation domain space was filled up with 275 argon vapor atoms. Both the copper and argon atoms were arranged in an fcc (100) lattice. Surface modification was made by adding four copper nanostructures; each nanostructure was of the cuboid shape of $0.58 \times 1.42 \times 0.58$ nm$^3$ and placed over the solid copper wall. For performing simulation with nano slots, three copper nano slots were created, each of which had a width of 0.58 nm and depth 0.58 nm. Surface modification was also made by creating a hemispherical void at the top layer of the solid wall. To examine the evaporation behavior from the same effective surface area, first, one nano structure was created on the solid surface. Then the nano structure was split into two nano structures in a way that total effective surface area remained the same. Figure 1 shows the initial simulation domain configuration and different modifications of solid surfaces employed in this study.

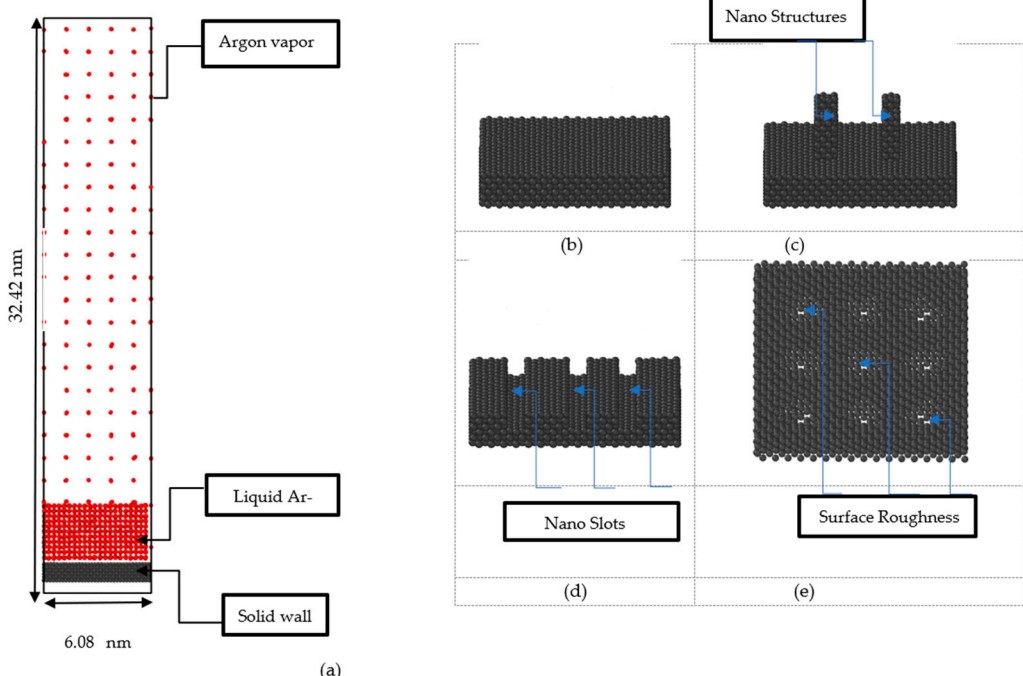

**Figure 1.** Simulation domain and solid wall configuration: (**a**) initial simulation domain for smooth surface. Different surface texture: (**b**) smooth surface, (**c**) surface with nano structures, (**d**) surface with nano slots, (**e**) surface with nano level roughness (to make the figure (**e**) clear, we changed the viewing angle and the radius of the particles).

For describing the inter-atomic interaction, 12-6 Lennard-Jones (LJ) potential [16] was used in this study as shown in Equation (1). The energy parameter ($\epsilon$) and the length parameter ($\sigma$) used in this simulation are: $\sigma_{Ar\text{-}Ar}$ = 0.34 nm and $\epsilon_{Ar\text{-}Ar}$ = 0.0104 eV [17]; $\sigma_{Cu\text{-}Cu}$ = 0.23 nm and $\epsilon_{Cu\text{-}Cu}$ = 0.415 eV [18]. The interaction parameters between copper and argon were calculated by using the Lorentz–Berthelot mixing rule [19] as shown in Equations (2) and (3).

In the case of different surface wetting conditions, the energy parameter found from the Lorentz–Berthelot mixing rule was multiplied by a wetting factor. For hydrophilic and hydrophobic conditions of the surface, the factor is taken as 1.0 and 0.1, respectively. To reduce the computational times, Morshed et al. [12] and Wang [13] et al. truncated all the potentials at 3.5 $\sigma_{Ar\text{-}Ar}$. Hence, the cut off radius was selected as 3.5 $\sigma_{Ar\text{-}Ar}$. This cut off radius is long enough to capture all the interactions of an atom with its surrounding atoms.

$$\Phi\left(r_{ij}\right) = 4\epsilon \left[ \left( \frac{\sigma}{r_{ij}} \right)^{12} - \left( \frac{\sigma}{r_{ij}} \right)^{6} \right] r_{ij} < r_{cutoff} \tag{1}$$

$$\sigma_{ij} = \frac{\sigma_{ii} + \sigma_{jj}}{2} \tag{2}$$

$$\epsilon_{ij} = \sqrt{\epsilon_{ii}\epsilon_{jj}} \tag{3}$$

where $r$ is the distance between two atoms, $\sigma$ is the characteristic length that is a finite distance at which the interparticle potential becomes zero, and $\varepsilon$ denotes the potential well depth [13]. The equation of motion was employed and integrated using the velocity Verlet algorithm. The time step used in this simulation was 5 fs. The periodic boundary condition was used in x and z directions whereas the fixed boundary condition was used in the y-direction. The periodic boundary condition is equivalent to considering infinite, space-filling arrays of identical copies of the simulation region where atoms can interact across the boundary; whereas in fixed boundary condition, atoms do not interact across the boundary and the position of the face is fixed. The Langevin thermostat was applied to the solid layers

and kept on at 90 K for 0.5 ns starting from the initial configuration of the simulation domain. The thermostat was then turned off and the molecular system was allowed to equilibrate for 0.5 ns more. The equilibrium condition was ensured by checking different thermodynamic states such as pressure, temperature, and density. After reaching equilibrium, to allow the phase transition of argon atoms, the Langevin thermostat was set to 140 K and simulation in this condition was run for 5 ns. To keep the simulation temperature far below the explosive boiling temperature, we tried to find the exact explosive boiling temperature of argon. The explosive boiling temperature of argon on copper plate was found at 173 K which has been showed in Figure 2 As all our simulations are done far below 173 K temperature, we avoided the explosive boiling zone of liquid argon.

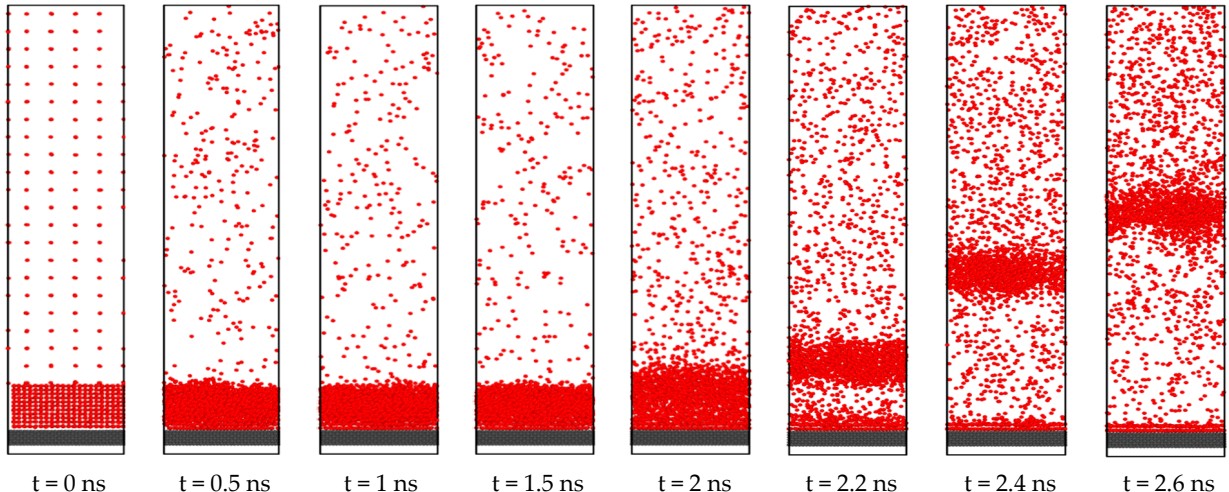

| t = 0 ns | t = 0.5 ns | t = 1 ns | t = 1.5 ns | t = 2 ns | t = 2.2 ns | t = 2.4 ns | t = 2.6 ns |

**Figure 2.** Explosive boiling of liquid argon at 173 K temperature.

*Reason for Using Liquid Argon and Extension to Complex Fluid*

Argon has been considered in this simulation because its interatomic potentials are well known, and these results can be extended to complex fluids like water. The simulation of homogenous boiling of water and heterogenous explosive boiling of water is shown in Figures 3 and 4. The homogenous boiling of water was done at a temperature of 400 K. From Figure 3, it is obvious that the boiling water shows similar behavior to liquid argon. The explosive boiling temperature of water was observed to be 500 K [15]. The main difference between the explosive boiling of argon and water is that, in explosive boiling of water all the particles leave the surface as a single chunk, whereas during the explosive boiling of argon, particles leave the surface both as a single chunk or multiple chunks and as some discrete particles. The lesser interatomic bonding between argon particles is responsible for the discrepancy in the behavior. Shahadat et al. showed that for water the amount of energy transfer is more for the hydrophilic surface than the hydrophobic surface, while for the hydrophilic surface the water is attracted to the solid surface more than the hydrophobic surface [15]. For liquid argon we found a similar trend.

From the simulation of both homogenous boiling and explosive boiling of water and from the previous literature, we can conclude that a complex fluid like water shows similar boiling patterns to liquid argon. Therefore, these results can be extended to more complex fluids than liquid argon.

In this research, the motivation was to observe the nature of heat transfer at evaporation or normal boiling stage. As the results from these simulations can be easily expanded to more complex fluids, only liquid argon has been considered in this study.

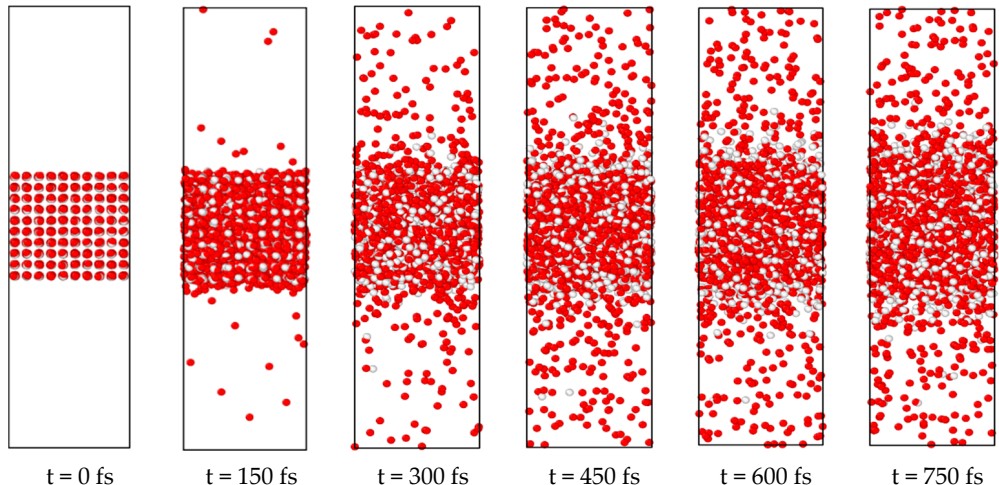

**Figure 3.** Homogenous boiling of water at 400 K temperature.

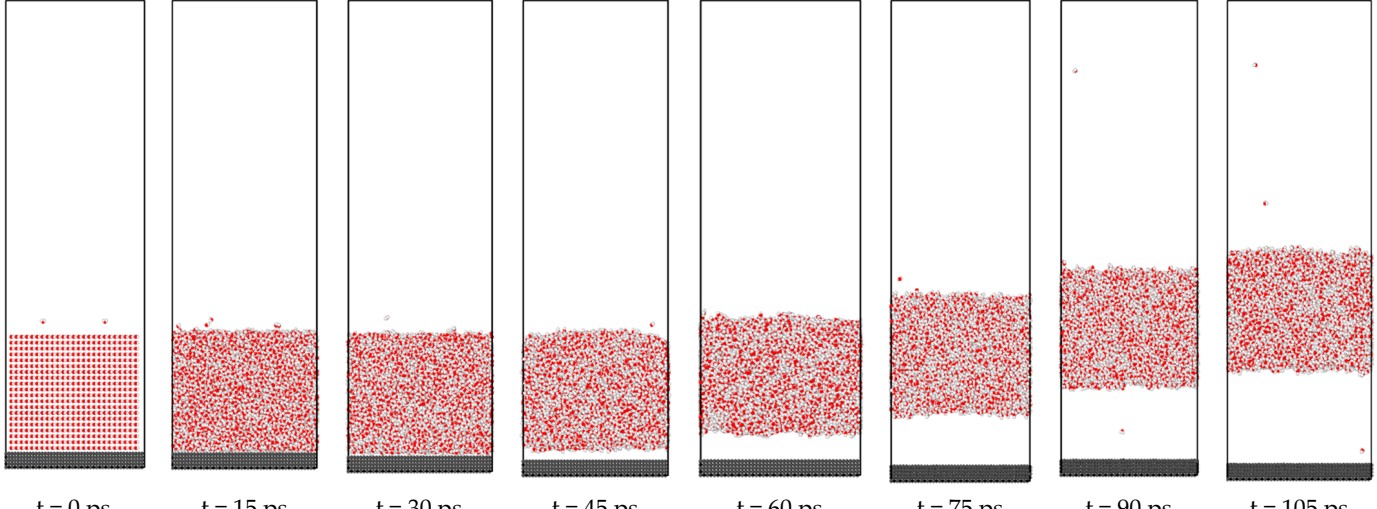

**Figure 4.** Explosive boiling of water at 500 K temperature.

All the simulations used in this study were performed by using LAMMPS (Large-scale Atomic/Molecular Massively Parallel Simulator) [20] and visualization was done by using OVITO [21]. LAMMPS is a classical molecular dynamics code with a focus on materials modeling. It is an acronym for large-scale atomic/molecular massively parallel simulator. LAMMPS has potential for solid-state materials (metals, semiconductors), soft matter (biomolecules, polymers), and coarse-grained or mesoscopic systems. It can be used to model atoms or, more generically, as a parallel particle simulator at the atomic, meso, or continuum scale [22]. OVITO (Open Visualization Tool) is a scientific visualization and analysis software for molecular and other particle-based datasets, typically generated by numeric simulation models in materials science, physics, and chemistry disciplines [21].

## 3. Results and Discussion

In this study, phase change heat transfer for different surface textures such as surface wetting condition, nanostructured surface, surface roughness, and surface with nano slot was investigated. Copper was used as a solid wall. Figure 5 shows, a solid wall temperature for different surface conditions. It shows that the solid wall temperature is almost identical in each case which suggests that surface textures have no effect on the temperature of the solid wall. Snapshots of the simulation domain, the density profile of argon atoms, and net evaporation number for different time steps were used to explain the results.

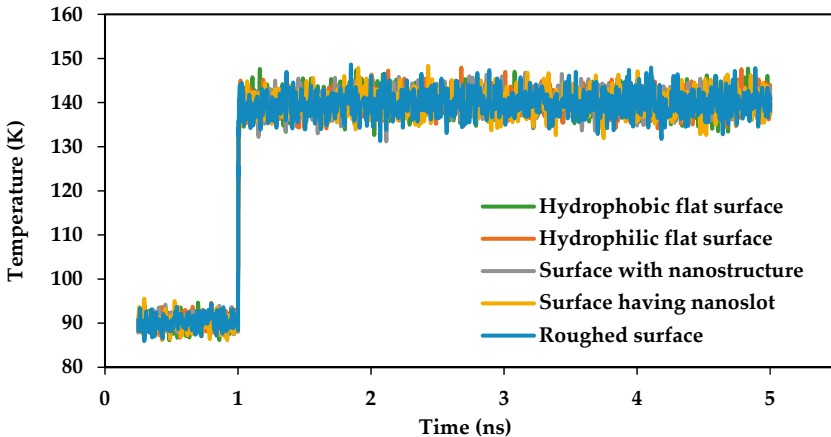

**Figure 5.** Temperature of solid wall for different surface textures.

### 3.1. Effect of Surface Wetting Condition

Heat transfer performance and phase change phenomena largely depend on the condition of the solid-liquid surface interface. The effect of surface interface condition on the boiling heat transfer and different solid-liquid surface wettability was considered. Five different surface wetting conditions were chosen in this study. Snapshots of simulation domain for different surface wettability at different times were shown in Figure 6. In the beginning, there is no movement of atoms from the liquid region to the vapor region. As time progresses, atoms from the liquid domain start to migrate to the vapor domain which means evaporation takes place. This movement of atoms increases when the surface wetting condition increases. At 2 ns, there are many argon atoms in the liquid domain for the hydrophobic surface (wetting factor = 0.1). When the wetting condition is gradually changed to hydrophilic (wetting factor = 0.5 and 1.0), more argon atoms from the liquid region move to the vapor region. The thickness of the liquid layer decreases at an earlier phase as the wettability increases. Hence, increasing the surface wettability ensures better heat transfer performance and therefore helps to enhance evaporation.

The temperature history of the argon atoms is shown in Figure 7. It is observed that argon atoms take more time to reach equilibrium with the solid wall temperature for the hydrophobic condition (wetting factor = 0.1). When the wetting condition changes, argon takes less time to reach the equilibrium condition with the solid wall. The rapid increase of temperature in the case of a hydrophilic condition (wetting factor = 1.0) indicates that energy is transferred more quickly from the solid wall to liquid due to better solid-liquid interaction. From the temperature history of argon atoms, it can easily be said that more heat transfer takes place where the surface is more hydrophilic. Hence, transforming the surface from hydrophobic to hydrophilic conditions ensures the enhancement of evaporation phenomena.

The density profile is an important parameter to explain the heat transfer performance where phase change takes place. Figure 8 shows the density profile of argon for different surface wetting conditions at 4 ns. From the figure, it is seen that near the solid wall the density of argon is higher for all the cases. In the case of the hydrophilic conditions, the density of argon is highest near the wall because of strong interaction with the solid wall. As the wettability decreases, interaction with the solid wall decreases and results in a decrease in the density of argon. At the liquid-vapor interface, the density of argon for wetting factor 0.1 is higher than in any other case. It indicates that it has a thicker non-evaporating layer and thus less heat transfer takes place. Figure 9 shows the net evaporation number with time steps for different surface wetting conditions. The change of evaporation number is almost linear up to 2.5 ns. In this linear region, the slope of the evaporation number-time curve for wetting factor 1.0 is more than any other case. This indicates that evaporation occurs at a quicker rate for the wetting factor 1.0. Therefore, by

increasing the surface wetting condition, the heat transfer rate can be enhanced, and the enhancement of evaporation takes place.

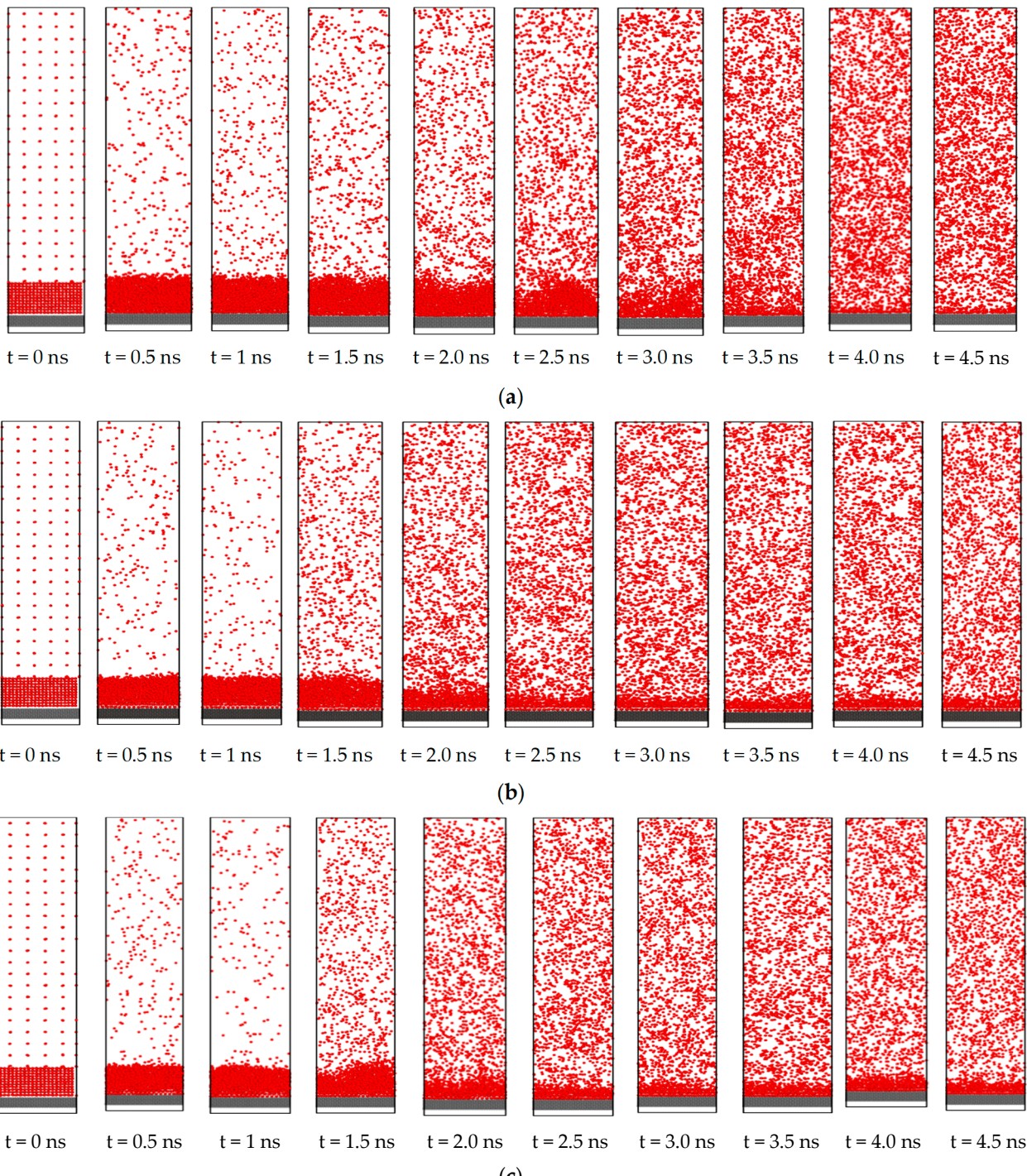

**Figure 6.** Simulation snapshots in the (x-y) plane for different wetting conditions. Wetting factor = 0.1 (**a**), 0.5 (**b**), 1.0 (**c**).

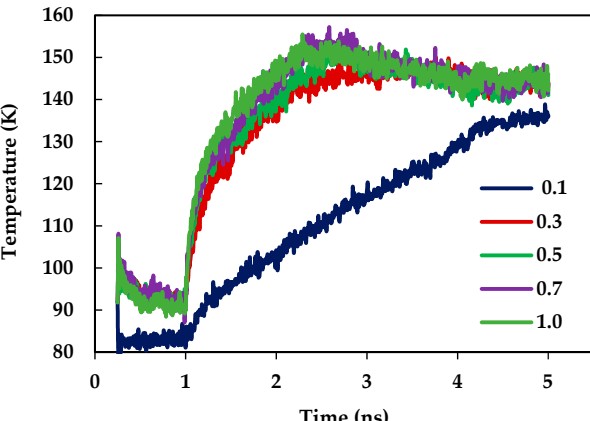

**Figure 7.** Temperature history of argon atoms for different surface wetting conditions.

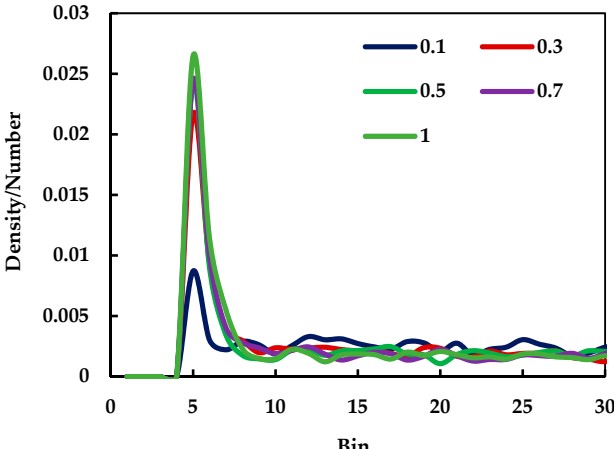

**Figure 8.** Density profile of argon at 4 ns for different surface wetting conditions.

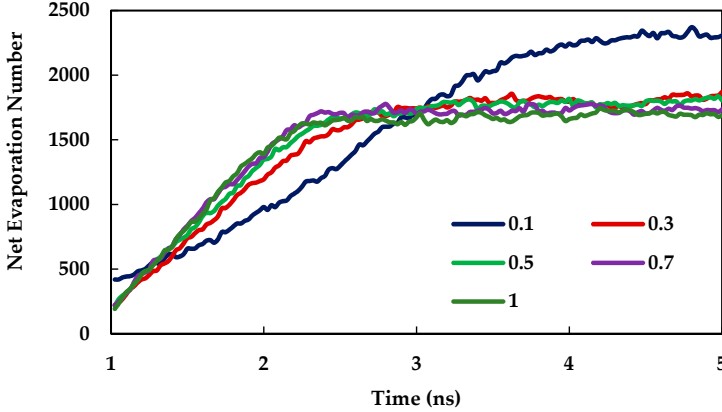

**Figure 9.** Net evaporation number for different surface wettability as a function of time.

### 3.2. Effect of Nano Structures

The second modification was to introduce nano structures on the surface. Introducing nano structures results in heat transfer enhancement during phase change phenomena because heat transfer increases with the increase of surface area.

When nanostructure is introduced, more liquid argon particles get closer to the solid surface and can take heat easily from the wall and become vapor. In this study, four nanostructures are placed on the solid surface. It is clear from Figure 10a that evaporation

begins from the upper layer of liquid argon. Molecules of liquid argon escape into the vapor region from the top layer as individual molecules or tiny clusters. The presence of nanostructure increases the solid-liquid interface area and interaction, which results in a faster energy transfer from the solid wall to the liquid molecules. Introducing these four nanostructures increases the total surface area by 43.36 nm$^2$. This huge increase in surface area results in the enhancement of evaporation from the solid wall.

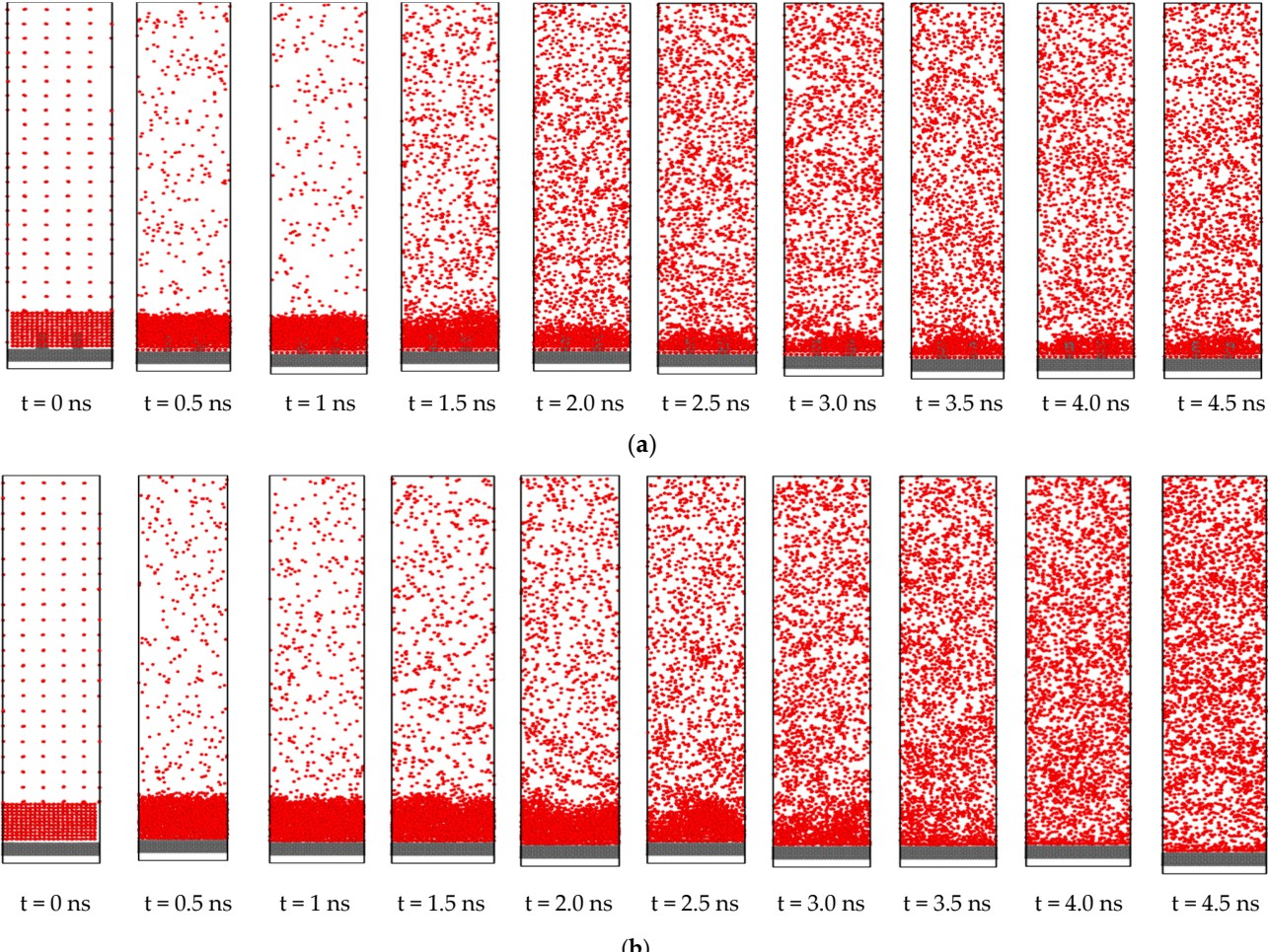

**Figure 10.** Snapshots from the simulation domain (*x*–*y* plane) for (**a**) surface with nano structure (**b**) smooth surface.

The temperature history of the argon atoms is shown in Figure 11. The rapid increase of temperature for nanostructures indicates that energy is transferred more quickly from the solid wall to liquid due to increased surface area and better solid-liquid interaction. From the temperature history of the argon atoms, it can easily be said that more heat transfer takes place when nanostructures are added. Hence, introducing nanostructures at the solid surface ensures the enhancement of evaporation.

Figure 12 shows the density profile of argon for the condition with nanostructures at 4 ns. From the figure, it can be seen that near the solid wall the density of argon is higher. In the case of nanostructures, the density of argon is highest near the wall because of strong interaction with the solid wall. For the solid wall without nanostructures the interaction of the liquid with the solid wall decreases. Therefore, the argon density decreases near the wall.

Figure 13 shows the net evaporation number with time for the case of nanostructures and without nanostructures. The change of evaporation number is almost linear up to 2 ns. In this linear region, the slope of the evaporation number-time curve for nanostructure is more than for the solid wall. This indicates that evaporation occurs at a quicker rate when

nanostructures are introduced. Therefore, by placing nanostructures on the solid surface, the heat transfer rate can be enhanced, and enhancement of evaporation takes place.

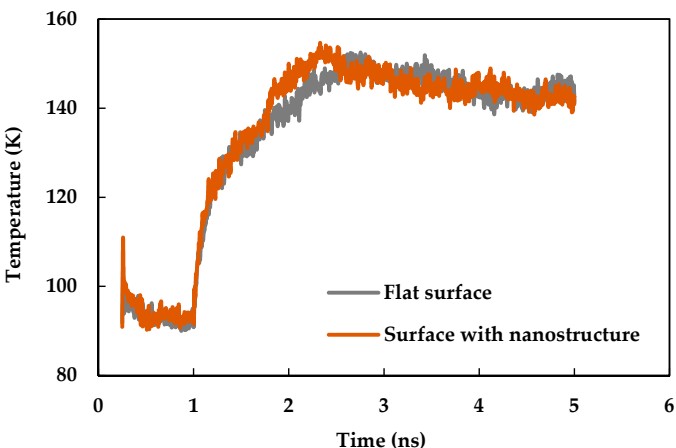

**Figure 11.** Comparison of temperature history of argon atoms for nano-structured surface with smooth surface.

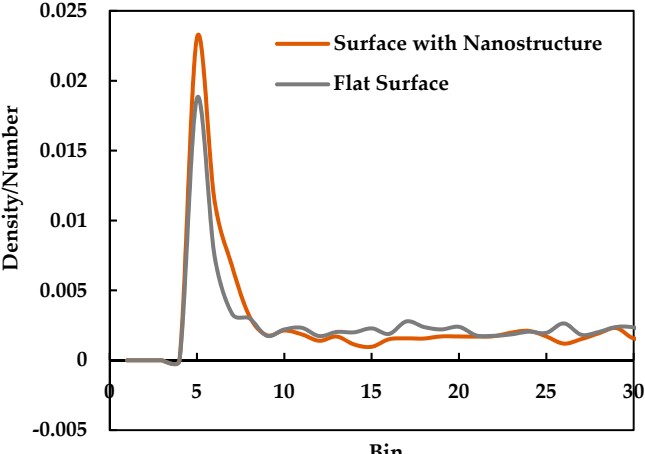

**Figure 12.** Comparison of density profile of argon at 4 ns for nano-structured surface with smooth surface.

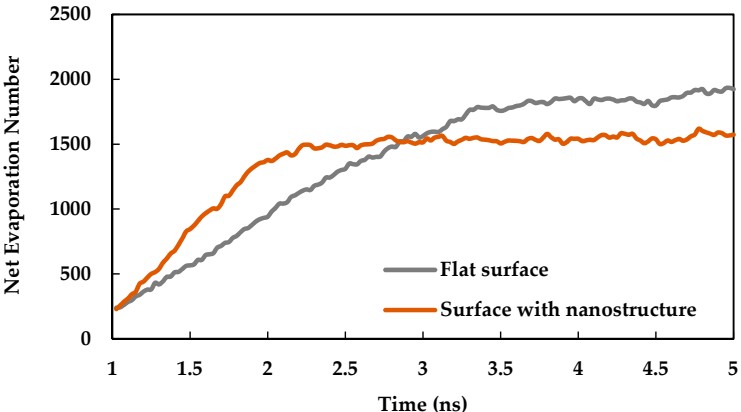

**Figure 13.** Net evaporation number as a function of time.

### 3.3. Effect of Nano Slots

The third surface modification was to introduce some nano slots on the smooth copper surface. If any nano level slot is cut from the solid surface, it will alter the surface texture

and the morphology and therefore it will change the heat transfer behavior from that surface. In this study, to examine the effect of nano slots on the rate of evaporation, three nano slots were created on the solid copper surface and then the results were compared with the smooth solid surface. The solid wall was subjected to a heating of 140 K which was just above the boiling point of the adjacent liquid and heat flowed from the solid surface to the liquid. As the temperature was high enough, the evaporation of liquid argon from the copper wall began. Figure 14 shows the snapshots of the simulation domain for the surface having nano slots.

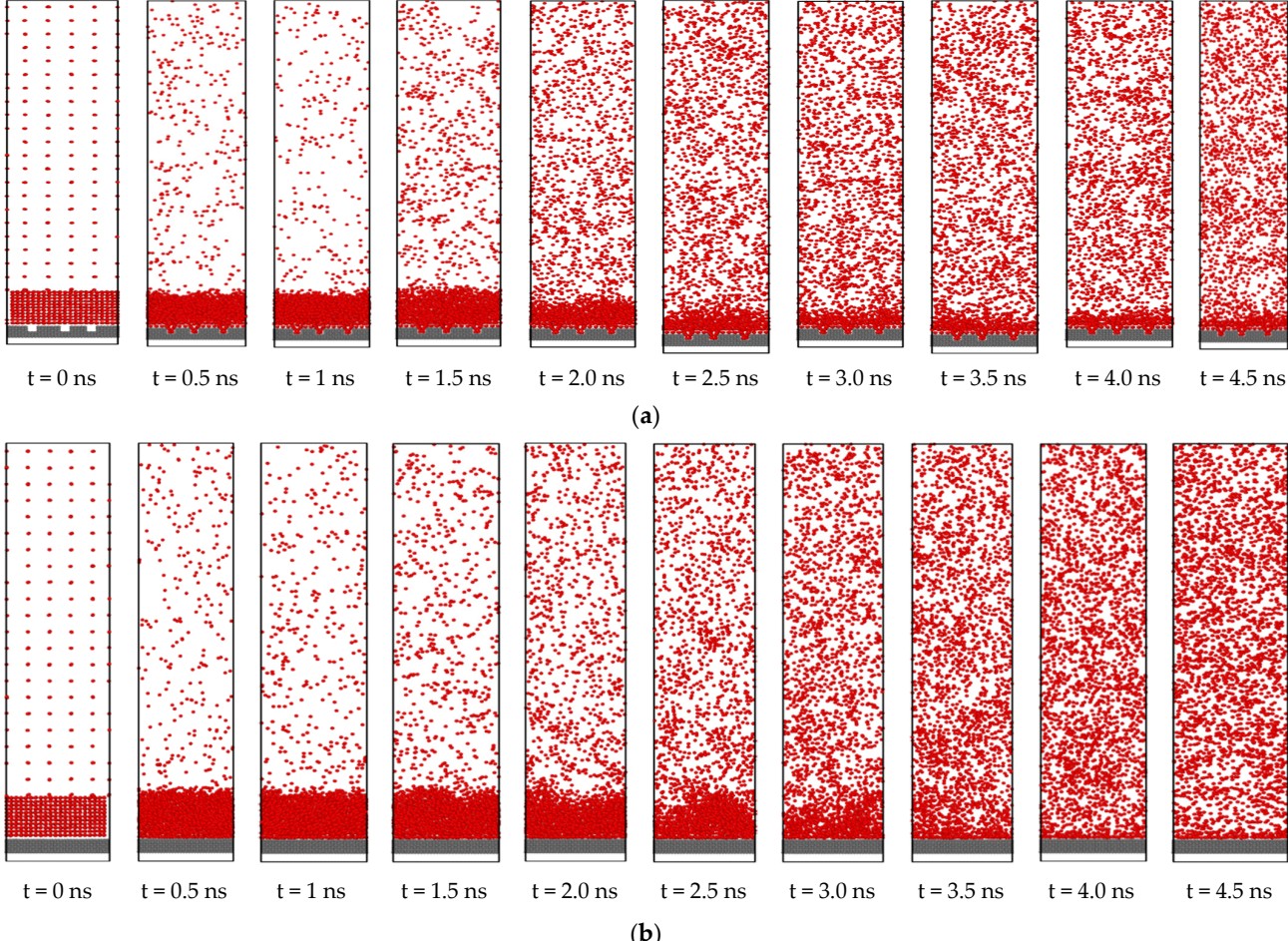

**Figure 14.** Snapshots from the simulation domain (*x–y* plane) for (**a**) surface having nano slots, (**b**) smooth surface.

The temperature history of the argon atoms is shown in Figure 15. It is observed that the argon atoms take more time to reach to equilibrium with the solid wall without nano slots. The rapid increase of temperature for nano slots indicates that energy is transferred more quickly from the solid wall to liquid due to increased surface area and better solid-liquid interaction. From the temperature history of the argon atoms, it can easily be said that more heat transfer takes place when nano slots are added.

Figure 16 shows the density profile of argon for the condition with nano slots at 4 ns. It is observed that in the case of nano slots, the density of argon is highest near the wall because of strong interaction with the solid wall. For a solid wall without nano slots, the interaction of liquid with the solid wall decreases and that is why near the wall, the density of argon also decreases. Figure 17 shows the net evaporation number with time steps for the case of nano slots and without nano slots. The rate of evaporation was basically determined by considering the number of liquid particles transferring into the vapor zone in a certain time step. From the figure, it is evident that the initial rate of evaporation is higher for the surface with the nano slots than the smooth surface. This means the

evaporation takes place at a faster rate than the smooth surface. The reason behind this enhancement of the evaporation is that, when a nano slot is created, the effective surface area increases. In this case, nano slots increase the surface area by 63 nm$^2$. This increase in effective area results in the enhancement of the rate of evaporation from the smooth surface. In addition, due to the presence of the nano slot, the surface wettability increases, solid-liquid contact area increases, and the interfacial thermal resistance decreases which eventually increases the rate of evaporation. From Figure 13, it is also evident that the evaporation rate decreases earlier for the surface with nano slots compared to the smooth surface, which implies that liquid molecules have been evaporated earlier for the surface with nano slots.

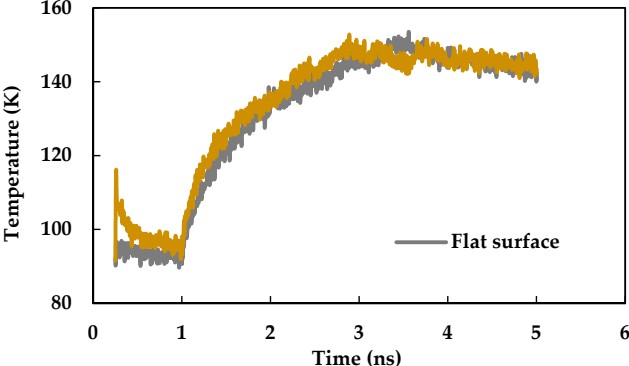

**Figure 15.** Comparison of temperature history of the argon atoms for a surface having nano slots with a smooth surface.

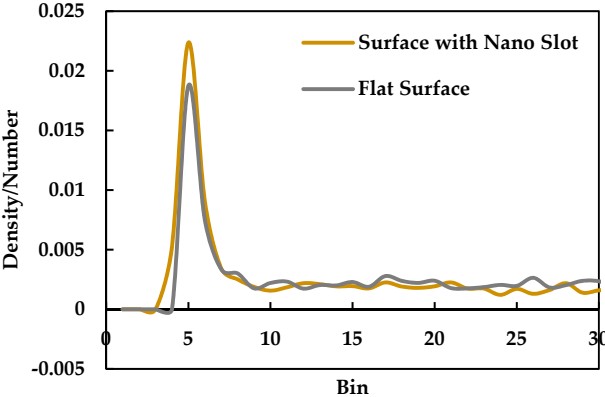

**Figure 16.** Comparison of density profile of argon at 4 ns for surface having nano slots with a smooth surface.

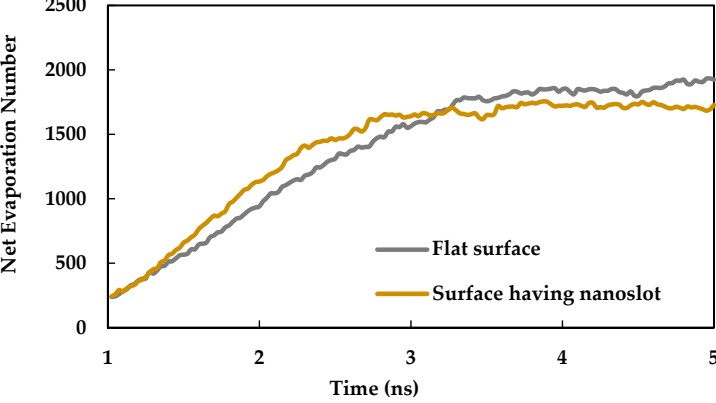

**Figure 17.** Net evaporation number as a function of time.

### 3.4. Effect of Nano Level Surface Roughness

The fourth type of surface modification is introducing nano-level surface roughness to the smooth surface. In this simulation, hemispherical surface roughness has been created by deleting some atoms from the smooth surface randomly. Creating this surface roughness also changes the surface texture and morphology which results in the modification of thermal transportation behavior. When surface roughness is created, it increases the effective heat transfer area which plays an important role in the enhancement of evaporation.

From the snapshot of the domain in Figure 18 and temperature history of the argon atoms shown in Figure 19, it is observed that the argon atoms take more time to reach equilibrium with the solid wall without surface roughness. When surface roughness is introduced to the solid surface, there is a very rapid growth of temperature. This rapid growth of temperature for surface roughness indicates that energy is transferred more quickly from the solid wall to liquid due to increased surface area and better solid-liquid interaction. Hence, it enhances the evaporation of liquid argon from the solid surface.

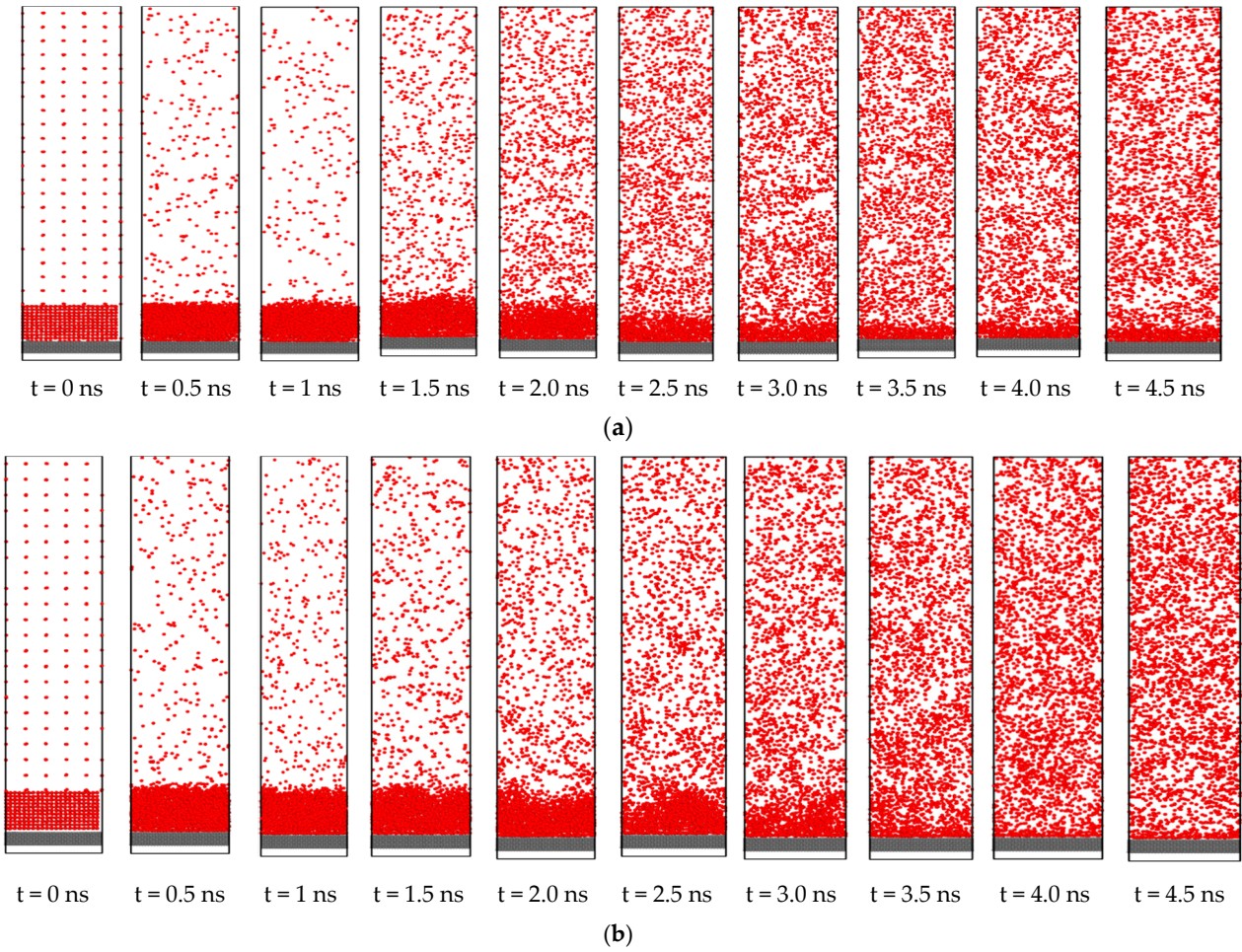

**Figure 18.** Snapshots from the simulation domain (*x–y* plane) for (**a**) surface with nano level roughness, (**b**) smooth surface.

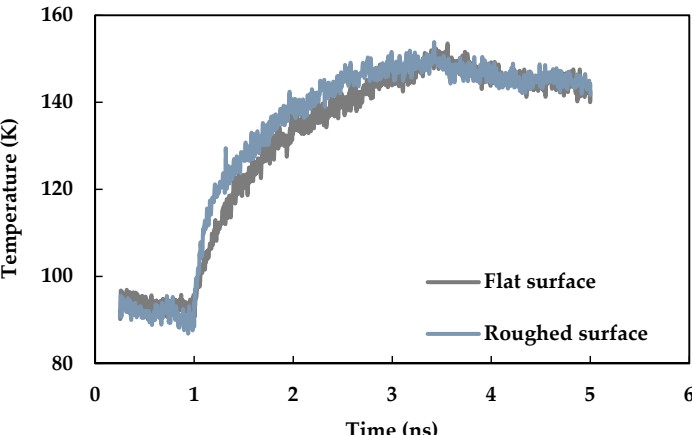

**Figure 19.** Comparison of temperature history of the argon atoms for nano level roughed surface with a smooth surface.

Figure 20 shows the density profile of argon for the condition with surface roughness at 4 ns. It is observed that in the case of surface roughness, the density of argon is highest near the wall because of strong interaction with the solid wall. From Figure 21, it is evident that the initial rate of evaporation is higher for the rough surface than the smooth surface. That means the evaporation takes place at a faster rate than from the smooth surface. The evaporation rate also decreases earlier for the rough surface compared to the smooth surface, which implies that liquid molecules have been evaporated earlier for the surface with roughness.

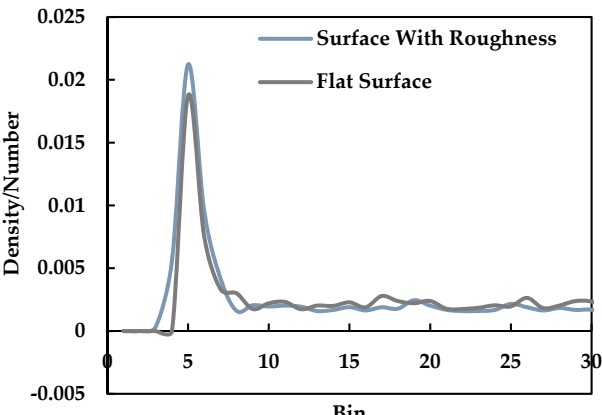

**Figure 20.** Comparison of density profile of argon at 4 ns for nano level roughed surface with a smooth surface.

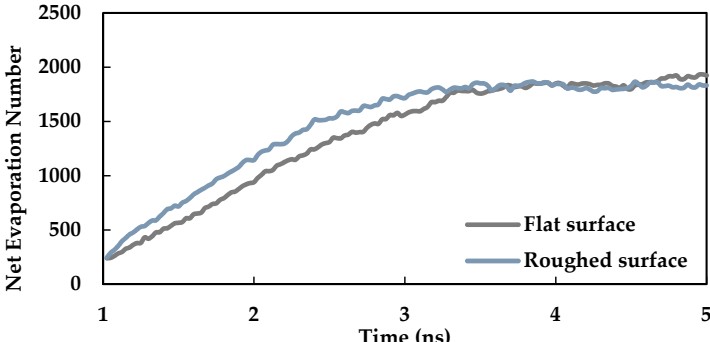

**Figure 21.** Net evaporation number as a function of time.

### 3.5. Effect of the Degree of Nano Structures for Same Effective Surface Area

To examine the evaporation behavior from the same effective surface area, two surface conditions were considered. First, one nano structure was created on the solid surface, then that one nano structure was split into two nano structures in a way that total effective surface area remained the same. This study will show whether there are some other factors affecting the thermal transportation behavior from the solid surface. Figure 22 shows the snapshots of the simulation domain for one nanostructure and two nanostructures.

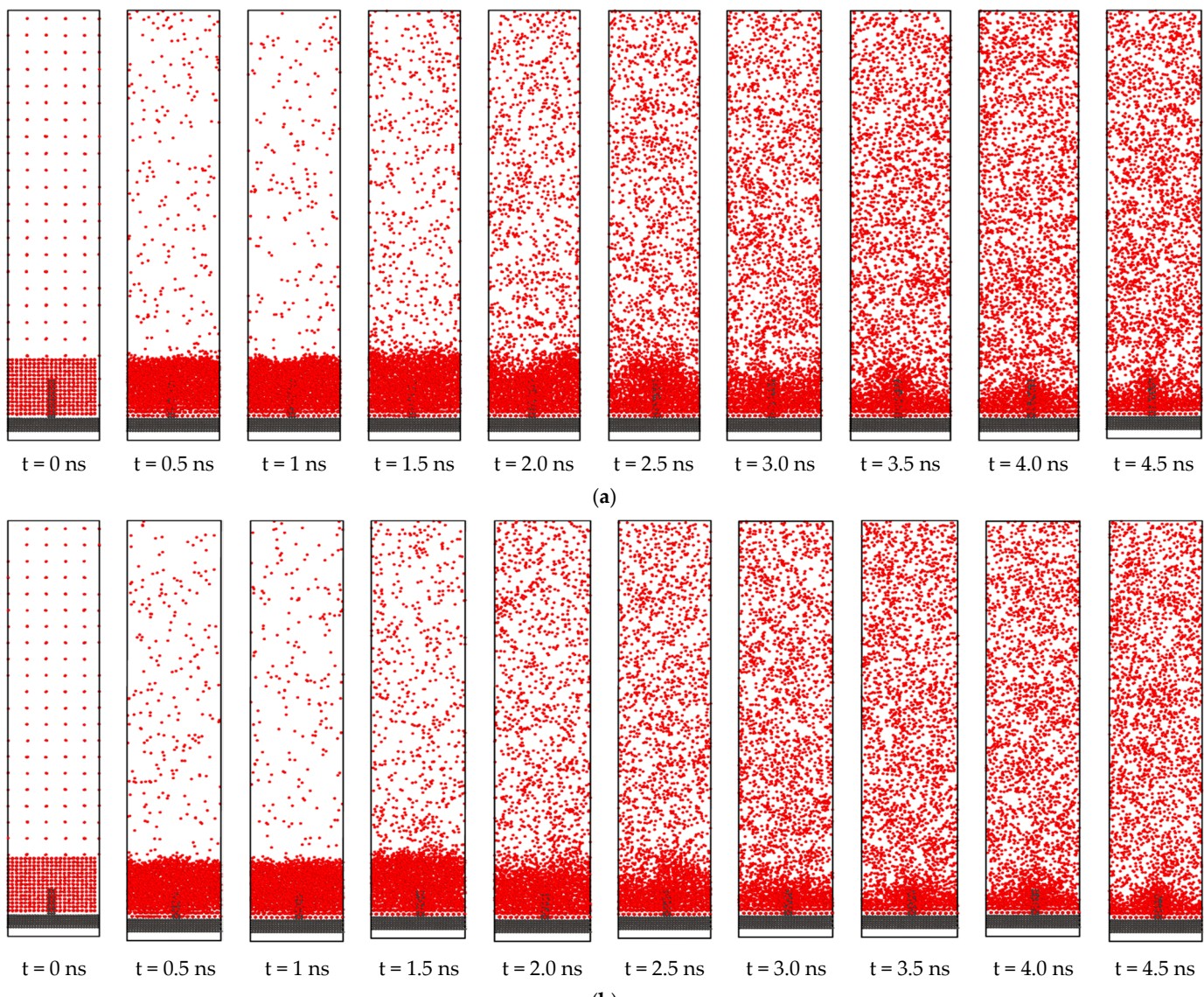

**Figure 22.** Snapshots from the simulation domain (*x*–*y* plane) for (**a**) one nanostructure, (**b**) two nano structures.

Figure 23 shows the density profile of argon at 4 ns. It is observed that for both the cases, the density of argon is almost the same near the wall because of the similar interaction with the solid wall. If we go with a very critical analysis, we can see that near the wall, the value of density for double nanostructures case is slightly higher. The difference is so small that we can eventually consider them as the same. From Figure 24, it is observed that the initial rate of evaporation is slightly higher for the double nanostructure case than the single nanostructure case. That means the evaporation takes place at a little faster rate than the single nanostructure case. This effect of the slight increase of the rate of evaporation for the double nanostructure case is due to increase of the active nucleation sites.

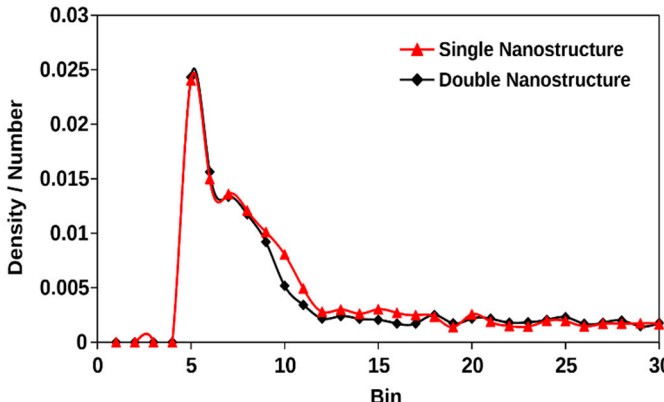

**Figure 23.** Comparison of density profile of argon at 4 ns for one nanostructure and two nanostructures.

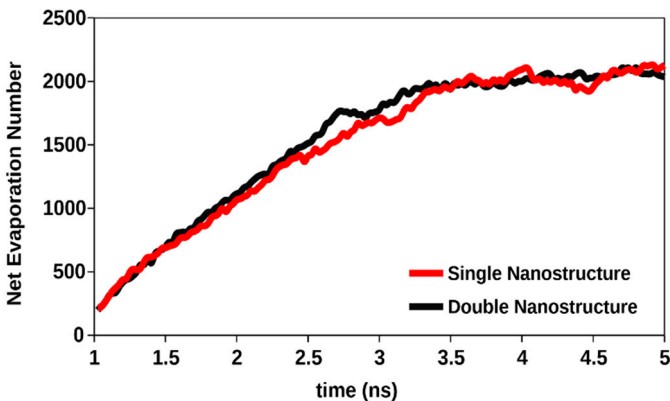

**Figure 24.** Net evaporation number as a function of time.

## 4. Conclusions

Nonequilibrium molecular dynamics simulation has been employed to study the enhancement of heat transfer in phase change phenomena. Four types of surface modification were considered: (i) transformation of hydrophobic to hydrophilic condition, (ii) introducing nano structures to the surface, (iii) cutting nano slots, (iv) introducing nano level surface roughness. In addition, heat transfer effects on the surfaces with the same effective surface area but a different number of nanostructures were studied. The following conclusions may be drawn:

- When the solid surface is transformed from hydrophobic condition to hydrophilic condition, an increase of heat transfer is observed during phase change phenomena. This happens due to the increase in the surface wettability condition. Increased surface wettability allows the liquid atoms to be more adjacent to the solid surface. Therefore, solid-liquid interaction is modified, and it results in an enhancement of heat transfer.
- The heat transfer rate from solid surface to liquid is enhanced for a nanostructured surface compared to a flat surface due to the increase of the effective surface area.
- Introducing nano slots increases the effective surface area as well as solid-liquid interaction and decreases the resistance to heat flow. Therefore, it results in an enhancement of heat transfer during phase change phenomena.
- Nano level surface roughness enhances heat transfer during phase change by increasing the effective surface area and solid-liquid interaction and decreasing the interfacial thermal resistance.
- If the effective surface area is kept the same, heat transfer during phase change can still be very slightly modified by increasing the number of active nucleation sites. When two nanostructures were introduced on the solid surface instead of one, keeping

the effective surface area the same, there was still a very slight increase of the net evaporation number.

**Author Contributions:** Conceptualization, methodology, software, and formal analysis: M.R.B.S., M.R.H.R., A.S.M., S.A.-A.S.; writing—original draft preparation: M.R.B.S., M.R.H.R., A.S.M., S.A.-A.S.; review and editing: M.R.B.S., M.R.H.R., A.S.M., S.A.-A.S., A.K.M.M.M., T.C.P.; supervision: A.K.M.M.M., T.C.P. All authors have read and agreed to the published version of the manuscript.

**Funding:** This research received no external funding.

**Institutional Review Board Statement:** This study did not involve humans or animals.

**Informed Consent Statement:** This study did not involve any human.

**Data Availability Statement:** This study did not report any data.

**Acknowledgments:** Authors would like to thank the Department of Mechanical Engineering at Bangladesh University of Engineering and Technology (BUET) for providing technical supports for this project.

**Conflicts of Interest:** The authors declare no conflict of interest.

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
