# Peer review of "A Molecular Dynamics Study of Heat Transfer Enhancement during Phase Change from a Nanoengineered Solid Surface"

_processes, doi:10.3390/pr9040715_

Round 1
Reviewer 1 Report
Review comments for Processes-1160303
The authors present an interesting study of heat transfer enhancement during phase change from a nano engineered surface using molecular dynamics. Language errors should be corrected (the articles “the” and “a” are missing in places or misused in others). It would also be good to improve the narrative and clarity of the text. There are a few technical deficiencies to be addressed. Herewith some improving suggestions:
- Page 1-Title: Write “of Heat Transfer Enhancement” instead of “of Enhancement of Heat Transfer”.
- P1: Correct language errors and improve the narrative and clarity of the Abstract.
- P1-2: Correct language errors and improve the narrative and clarity of the Introduction.
- P2-4: Correct language errors and improve the narrative and clarity of section 2.
- P3: Explain properly the symbols in equations (1) to (3).
- P5: Improve the clarity of the caption of Fig. 3.
- P5: Although references are cited for LAMMPS and OVITO, it would be good for the novice reader to provide brief descriptions.
- P6: Improve the clarity of Fig. 5.
- P6-9: Correct language errors and improve the narrative and clarity of section 3.1.
- P7: Simplify the caption of Fig. 6 to read: “Simulation snapshots in the (x-y) plane for different wetting conditions. Wetting factor= 0.1 (a), 0.5 (b), 1.0 (c)”.
- P8: 08. P6: Improve the clarity of Fig. 7.
- P9-11: Correct language errors and improve the narrative and clarity of section 3.2.
- P10: Improve the clarity of Fig. 11.
- P11: Correct language errors and improve the narrative and clarity of section 3.3.
- P12: Improve the clarity of Fig. 15.
- P13: Correct language errors and improve the narrative and clarity of section 3.4.
- P14: Improve the clarity of Fig. 19.
- P15: Correct language errors and improve the narrative and clarity of the Conclusions.
- P16: Delete texts of “Authors Contributions” and “Funding” as redundant.
- List references complete, correctly, consistently and according to the standards of the Journal.
Author Response
We would like to thank the reviewers for their time, effort, and dedication in providing us valuable feedback on our manuscript. We are grateful to the reviewers for their insightful comments on our paper. We have been able to incorporate changes to reflect all the suggestions provided by the reviewers. We have also highlighted the changes within the manuscript.
Here is a point-by-point response to the reviewers’ comments and concerns.
Comment 1. Line 15: 1. Page 1-Title: Write “of Heat Transfer Enhancement” instead of “of Enhancement of Heat Transfer”.
Response: Thank you for pointing this out. The name of the manuscript has been edited accordingly.
Comment 2. P1: Correct language errors and improve the narrative and clarity of the Abstract.
Response: Thank you for your valuable suggestion. The abstract has been edited accordingly.
Comment 3. P1-2: Correct language errors and improve the narrative and clarity of the Introduction.
Response: Thank you for your valuable suggestion. The Introduction has been edited accordingly.
Comment 4. P2-4: Correct language errors and improve the narrative and clarity of section 2.
Response: Thank you for pointing this out. The manuscript has been edited accordingly.
Comment 5. P3: Explain properly the symbols in equations (1) to (3).
Response: Thank you so much for your suggestion. The symbols have been explained in the manuscript.
|
|
(1) |
|
(2) |
|
|
|
|
|
(3) |
|
|
|
|
where r is the distance between two atoms, σ is the characteristic length that is a finite distance at which the interparticle potential becomes zero, and ε denotes the potential well depth.
Comment 6. P5: Improve the clarity of the caption of Fig. 3.
Response: Thank you for your valuable suggestion. The caption has been modified.
Comment 7. P5: Although references are cited for LAMMPS and OVITO, it would be good for the novice reader to provide brief descriptions.
Response: Thank you so much for your suggestion. The following details have been included in the manuscript.
“All the simulations used in this study were performed by using LAMMPS [1] and visualization was done by using OVITO [2]. LAMMPS is a classical molecular dynamics code with a focus on materials modeling. It's an acronym for Large-scale Atomic/Molecular Massively Parallel Simulator. LAMMPS has potentials for solid-state materials (metals, semiconductors) and soft matter (biomolecules, polymers) and coarse-grained or mesoscopic systems. It can be used to model atoms or, more generically, as a parallel particle simulator at the atomic, meso, or continuum scale [3]. OVITO is a scientific visualization and analysis software for molecular and other particle-based dataset, typically generated by numeric simulation models in materials science, physics and chemistry disciplines [2].”
Reference
- Plimpton, S. Computational limits of classical molecular dynamics simulations. Computational Materials Science 1995, 4(4), 361-364.
- Stukowski, A. Visualization and analysis of atomistic simulation data with OVITO–the Open Visualization Tool. Modelling and Simulation in Materials Science and Engineering 2009,18(1), 015012.
- S. Plimpton, Fast Parallel Algorithms for Short-Range Molecular Dynamics, Journal of Computational Physics, 1995, 117, 1-19.
Comment 8. P6: Improve the clarity of Fig. 5.
Response: Thank you for your valuable suggestion. The figure 5 has been updated in the manuscript.
Comment 9. P6-9: Correct language errors and improve the narrative and clarity of section 3.1.
Response: Thank you for your valuable suggestion. The manuscript has been edited accordingly.
Comment 10. P7: Simplify the caption of Fig. 6 to read: “Simulation snapshots in the (x-y) plane for different wetting conditions. Wetting factor= 0.1 (a), 0.5 (b), 1.0 (c)”.
Response: Thank you for your valuable suggestion. The caption has been edited accordingly.
Comment 11. P8: 08. P6: Improve the clarity of Fig. 7.
Response: Thank you so much for your suggestion. The figure 7 has been updated in the manuscript.
Comment 12. P9-11: Correct language errors and improve the narrative and clarity of section 3.2.
Response: Thank you for pointing this out. The manuscript has been edited as per recommendation.
Comment 13. P10: Improve the clarity of Fig. 11.
Response: Thank you so much for your suggestion. The figure 11 has been updated in the manuscript.
Comment 14. P11: Correct language errors and improve the narrative and clarity of section 3.3.
Response: Thank you for pointing this out. The manuscript has been edited as per recommendation.
Comment 15. P12: Improve the clarity of Fig. 15.
Response: Thank you so much for your suggestion. The figure 15 has been updated in the manuscript.
Comment 16. P13: Correct language errors and improve the narrative and clarity of section 3.4.
Response: Thank you for pointing this out. The manuscript has been edited as per recommendation.
Comment 17. P14: Improve the clarity of Fig. 19.
Response: Thank you so much for your suggestion. The figure 19 has been updated in the manuscript.
Comment 18. P15: Correct language errors and improve the narrative and clarity of the Conclusions.
Response: Thank you so much for your suggestion. The conclusion has been modified as suggested.
Comment 19. P16: Delete texts of “Authors Contributions” and “Funding” as redundant.
Response: Thank you for your suggestion. Mentioned segments have been deleted from the manuscript.
Comment 20. List references complete, correctly, consistently and according to the standards of the Journal.
Response: Thank you so much for your suggestion. The reference section has been reviewed and modified accordingly in the manuscript.

Reviewer 2 Report
The authors have attempted to address most my previous comments and I recognise the effort they have put into this. However, there are still things that have not been addressed adequately.
Region of thermostat application is still not clear. Is it only at Argon atoms, only at copper atoms or both? This has to be clarified.
Only truncating the potential is not the recommended in the literature approach, especially for long range interactions.
The "roughness" in figure 1e is still not very clear.
Language and writing style need considerable improvement. I would recommend the authors to consult a native English speaker or colleague with more publishing experience or a professional proof reader.
The response to my last comment show that the results and current analysis show just an expected behaviour, meaning increasing the effective surface area would increase the heat transfer rate. It would be much more interesting and novel if the authors could show and evaluate the heat transfer effect on surfaces with the same surface area but with different degrees of roughness and different shapes.
Unfortunately, I still cannot identify much novelty in this manuscript.
Author Response
We would like to thank the reviewers for their time, effort, and dedication in providing us valuable feedback on our manuscript. We are grateful to the reviewers for their insightful comments on our paper. We have been able to incorporate changes to reflect all the suggestions provided by the reviewers. We have also highlighted the changes within the manuscript.
Here is a point-by-point response to the reviewers’ comments and concerns.

Round 2
Reviewer 1 Report
Processes-1160303:
The authors present an interesting study of heat transfer enhancement during phase change from a nano engineered surface using molecular dynamics. Language errors should be corrected (the articles “the” and “a” are missing in places or misused in others). It would also be good to improve the narrative and clarity of the text. There are a few technical deficiencies to be addressed. Herewith some improving suggestions:
- Page 1-Title: Write “of Heat Transfer Enhancement” instead of “of Enhancement of Heat Transfer”.
- P1: Correct language errors and improve the narrative and clarity of the Abstract.
- P1-2: Correct language errors and improve the narrative and clarity of the Introduction.
- P2-4: Correct language errors and improve the narrative and clarity of section 2.
- P3: Explain properly the symbols in equations (1) to (3).
- P5: Improve the clarity of the caption of Fig. 3.
- P5: Although references are cited for LAMMPS and OVITO, it would be good for the novice reader to provide brief descriptions.
- P6: Improve the clarity of Fig. 5.
- P6-9: Correct language errors and improve the narrative and clarity of section 3.1.
- P7: Simplify the caption of Fig. 6 to read: “Simulation snapshots in the (x-y) plane for different wetting conditions. Wetting factor= 0.1 (a), 0.5 (b), 1.0 (c)”.
- P8: 08. P6: Improve the clarity of Fig. 7.
- P9-11: Correct language errors and improve the narrative and clarity of section 3.2.
- P10: Improve the clarity of Fig. 11.
- P11: Correct language errors and improve the narrative and clarity of section 3.3.
- P12: Improve the clarity of Fig. 15.
- P13: Correct language errors and improve the narrative and clarity of section 3.4.
- P14: Improve the clarity of Fig. 19.
- P15: Correct language errors and improve the narrative and clarity of the Conclusions.
- P16: Delete texts of “Authors Contributions” and “Funding” as redundant.
- List references complete, correctly, consistently and according to the standards of the Journal.
Author Response
We would like to thank the reviewers for their time, effort, and dedication in providing us valuable feedback on our manuscript. We are grateful to the reviewers for their insightful comments on our paper. We have been able to incorporate changes to reflect all the suggestions provided by the reviewers. We have also highlighted the changes within the manuscript.

Reviewer 2 Report
I would like to thank the authors for taking into account all the comments and suggestions. No further comments.
Author Response
We would like to thank the reviewers for their time, effort, and dedication in providing us valuable feedback on our manuscript. We are grateful to the reviewers for their insightful comments on our paper.